# Urban Echoes: Exploring the Dynamic Realities of Cities through Digital Twins

Piero Boccardo , Luigi La Riccia * and Yogender Yadav

Inter-University Department of Regional & Urban Studies and Planning (DIST), Politecnico di Torino, 10125 Torino, Italy; piero.boccardo@polito.it (P.B.); yogender.yadav@polito.it (Y.Y.)
* Correspondence: luigi.lariccia@polito.it; Tel.: +39-0110905696

**Abstract:** Digital twins (DTs) are highly valuable tools for urban planning as they provide a virtual replica of the physical city, integrating real-time data and simulations to enhance the decision-making and management processes. The use of DTs expands the possibilities for data integration and visualization in urban contexts. This includes real-time data measurements from multiple sources, such as sensors and IoT devices, facilitating comprehensive insights. DT's virtual representation helps authorities and planners visualize urban dynamics and improve their understanding of urban ecosystems, energy efficiency, traffic management, emergency response, and more. DT supports the simulation and modeling of different scenarios in an urban built environment, enabling the predictive analysis of transformation decisions and the anticipation of future trends and challenges. This paper highlights the assumptions and ongoing progress in the development of a DT for the city of Turin (Italy), focusing on a range of applications, such as the extraction of built and natural environment features, land use data, road network and pavement quality, and signage, along with continuous model updates over time.

**Keywords:** 3D model; digital twins; urban planning; smart city; sustainable city

## 1. Introduction

The field of spatial planning is currently facing a number of new challenges, particularly in terms of the analysis, evaluation, and management of cities and territories. These challenges are a result of a number of factors, including the occurrence of extreme environmental emergencies, the effects of climate change, and the ongoing process of urbanization. With the number of people living in cities expected to reach 5 billion by 2030, it is becoming increasingly important for planning to be effective, intelligent, and open to innovation, both in terms of the tools and data that are used and the way in which these data are shared. These aspects are particularly highlighted by the United Nations Sustainable Development Goal 11, which states that it is necessary to "make cities inclusive, safe, resilient and sustainable" [1].

Today, the awareness of sustainability is profoundly changing the way in which the city should be managed. A smart city is a city that combines ICT with physical infrastructure, architecture, urban equipment, and people to address economic, environmental, and social issues in an integrated way. However, working in an integrated way is not an easy goal; the perspective is not only that of the search for sustainability but above all, that of the competitiveness that cities must now pursue at a global level. Van Agtmael and Bakker, in their work entitled "The Smartest Places on Earth" [2], counter the recent consensus that the American and Northern European economies have lost their innovation initiative and competitive edge by referring to the so-called rustbelts, i.e., the former industrial areas that are now turning into brain belts, revitalized through the training of innovative centers, start-ups, and new collaborations with universities and oriented toward the search for a new model of economic and, at the same time, sustainable competitiveness.

However, not all cities should become the new Silicon Valley, and "Innovation isn't what you think" [3]; a misleading idea of innovation often prevails, which does not take into account the enormous variety of daily innovations that can lead to real progress and development, but which depend on the capacity of the context in which they are inserted, where it is necessary to rethink the role of local ecosystems and the importance of collaboration. Among the emerging innovations, digital twins (DTs) could constitute a significant technology because they represent dynamic and interactive virtual replicas of the physical environment of cities, starting from an extremely accurate 3D model in which near-to-real-time information is aggregated. The success of the application of DT will depend on the extent to which they can represent the complexity of the local ecosystems of reference.

The real potential of a DT is to have a 3D digital model that allows us to capture the entire city and determine real-world impacts to make timely decisions; to do so, the model must also be given workflows that use standardized metrics and processes to manage and maintain city elements and also to evaluate ongoing projects. The concept of 3D urban models has evolved, particularly with the integration of geographic information systems (GIS) and real-time data. This evolution is fundamental for analyzing dynamic urban elements, such as traffic patterns, environmental changes, and population dynamics. In addition, the potential of augmented reality (AR) and virtual reality (VR) in urban modeling offers immersive experiences that open new frontiers; 3D urban models have become increasingly important in urban planning because they enable better visualization of data, favor participatory planning and collaborative processes, and help in the simulation and evaluation of the impacts of urban transformation projects.

The field of spatial planning is increasingly reliant on the contributions of communities to spatial knowledge through online platforms, social media, and participatory GIS. This necessitates the treatment of spatial data as big and open data. This area of application, in conjunction with the integration of 3D modeling, represents a path of innovation within the spatial science disciplines. It is undoubtedly an attractive field for scientists and spatial planners who apply these theories in a wide range of disciplines. However, it would require the implementation of awareness-raising actions for the majority of users. The risk of DTs is, therefore, to foster critical glances between disciplinary fields that do not normally communicate and to allow city administrators to gain new perspectives on common frontiers. This is fundamental to an understanding of the realities and evolutionary processes of increasingly complex territorial systems in which institutional actors, economic actors, and environmental resources interact.

This paper aims to describe the progress and ongoing efforts in urban DTs, with a specific focus on the development of the Turin DT in Italy. The Turin DT project aims to create an advanced digital replica of the city, continuously updated with real-time data from internet of things (IoT) devices. In this paper, we have attempted to address and find solutions to the following research questions:

1.  What is the current state of the art in the field of city DTs in terms of practical implementations of DTs as a system?
2.  What are the strengths and weaknesses of urban digital twins, and how have the digital twins been used by cities for different purposes?
3.  With reference to the Turin DT project, what are the practical challenges faced in the implementation of digital twins for cities, and what are the possible solutions that have been explored so far?

## 2. Relevance of 3D City Models for Urban and Regional Planning

This section examines the evolution of 3D city models and their significance in the nascent field of geospatial sciences. It underscores the pivotal role of 3D city models in urban planning and regional development and how they serve as a valuable resource for various stakeholders engaged in planning and development processes. By offering detailed spatial representations, 3D city models facilitate comprehensive analysis and

decision-making in urban contexts. The integration of 3D city models into a digital twin (DT) enables planners, policymakers, and citizens to collaborate in an effective manner, thereby improving the efficiency and sustainability of urban development initiatives.

### 2.1. Technological Advancements in 3D City Models

The technological landscape of 3D city modeling has been reshaped significantly with advancements in GIS integration and real-time data assimilation. The fusion of GIS with 3D modeling [4] has been pivotal in enhancing the accuracy and relevance of urban models. This integration facilitates a multidimensional representation of urban spaces, encompassing topographical and infrastructural details. The introduction of real-time data into 3D urban models, considering the positive and negative aspects [5], has enabled dynamic simulation of urban environments. Although a digital 3D model of the city can be useful for visualizing important details and can help provide accurate information for the management of a smart city, the development and maintenance of a 3D model can indeed have some negative aspects, such as significant investment in terms of technology; technical complexity related to the use of a large amount of data; the need to change the organization of roles for managing the model within the public administration, etc.

Advances in remote sensing data collection techniques, such as light detection and ranging (LiDAR) and photogrammetry, have greatly improved the accuracy and efficiency of capturing urban environments. The integration of 3D city models with GIS platforms and semantic information (digital ontologies [6]) enables spatial analysis and the visualization of complex urban data. This integration also facilitates the incorporation of a metadata model that automates the labeling, classification, and analysis of objects and features, thereby promoting more intelligent applications, such as automated object recognition and urban monitoring.

This development is crucial for urban planning, allowing the analysis of traffic patterns, environmental changes, and population dynamics in real-time scenarios. The potential of AR and VR in urban modeling [7] offers immersive and interactive experiences, opening up new frontiers in urban planning and design, stakeholder engagement, tourism, and education.

### 2.2. Relevance of 3D City Models in Urban and Regional Planning

The existence of advanced data collection technology and sensors, particularly for environmental and traffic monitoring, is now mainstream and GIS must deal with near real-time data. New developments in visualization (geo-visualization) have transformed cartography by integrating animation, the search for salient features, and more. Similarly, the use of big data has led public administrations to consider new ways of using such data. In addition, the trend toward voluntary geographic information (VGI) and the willingness of people to participate in decision-making processes under the banner of crowdsourcing have implied the need to address these features efficiently.

Digital 3D city models have evolved significantly over time. Initially, they were simple three-dimensional representations used primarily for visualization. Over the years, they have become more complex and functional, integrating geospatial data and becoming essential tools for urban planning, resource management, and scenario simulation. Today, the most advanced techniques for building such city models include the photogrammetric management tools of remote sensing imagery (from aircraft or drones), combined with the availability of dense LiDAR point clouds to provide detailed and accurate measurements of the shape and physical characteristics of elements in the built and natural environment. There is also a need to integrate the potential of GIS with that of building information modeling (BIM) for architectural and environmental surveying.

The relevance of 3D city models in contemporary urban planning [8] lies in their ability to enhance the planning process in terms of data quality, technical complexity, scenario testing, and stakeholder engagement through visualization and communication. These models go beyond traditional 2D representations and provide stakeholders with a more

comprehensive understanding of proposed urban changes. Improved communication is necessary to promote collaborative planning processes. In addition, the efficiency that 3D modeling brings to urban planning has streamlined design processes, enabling rapid iterations and effective decision-making; GIS, which provides spatial data management and analysis capabilities, can be integrated with expert systems, which emulate human expertise in a particular domain, to support automated planning tasks.

The use of 3D city models in a DT plays a pivotal role in the field of sustainable urban development. These models provide invaluable tools for the simulation and evaluation of environmental impacts, thereby contributing to the creation of sustainable urban ecosystems. This is in accordance with the findings of [9], which highlight the significance of this approach. Furthermore, the application of these models in disaster management has been transformative, enabling the implementation of more effective resilience strategies [10]. This is an important area of planning research that uses spatial data and technology to mitigate risks, respond to crises, and aid recovery efforts. The utilization of interactive and realistic 3D models generated within the context of DT technology can facilitate the engagement of citizens in the planning process, thereby promoting a more inclusive approach to urban development [11]. This participatory planning ensures that urban development reflects the needs and aspirations of the community, which in turn leads to the creation of more liveable and equitable urban spaces. In this context, the democratization of urban planning facilitated by these models aligns with the principles outlined in Arnstein's work, "A Ladder of Citizen Participation" (1969) [12], and encourages inclusive and community-based urban development.

The advancements in urban 3D modeling technologies have enriched the processes of collaboration, sustainability, and public engagement in city development. These models stand as vital tools in the toolkit of modern urban planners, architects, and policymakers, driving the creation of more resilient, sustainable, and inclusive urban environments. By leveraging these new aspects, spatial planning can be more efficient, effective, and inclusive, leading to more sustainable, resilient, and livable communities.

## 3. Transition from 3D City Models to Urban DTs

The move from simple 3D models of cities to digital twins of cities marks a significant shift in the way that cities are understood and how they are managed. DTs, with their ability to use real-time data and advanced simulation capabilities, provide better insights for urban planning and decision-making. This transition is important for creating more resilient and sustainable cities by enabling more accurate predictions and responses to urban challenges.

### 3.1. Evolution of 3D City Models over Time

Three-dimensional city models generally have more advantages than two-dimensional maps, such as three-dimensional geometry of elements; light/shadow and lighting effects on elements; texture information; and the ability to change perspective for a correct view of the urban environment. At the same time, 3D city models also have some disadvantages, such as high production costs and the possibility that some models may be inaccurate at a certain level of detail. Until recently, the traditional method of 3D modeling required an enormous amount of manual work, including scanning a map to obtain a digital image, plotting the digital image with 3D CAD softwares, manual 3D modeling, extruding 2D contours to the height of the building, and/or modeling geometries from drawings and photographs.

Three-dimensional digital models of cities have traditionally been used to document urban assets and other dynamics to support urban planning activities; in general, such models represent physical objects statically, at a given point in time. This static state is clearly due to the sources used to generate these models, which "take a picture" at a precise point in time, such as aerial or drone remote sensing, nadir and oblique photography, and LiDAR point clouds. Updating the 3D model therefore often requires a new aerial

survey and a relative data processing time that takes a certain amount of time [13,14]. The transition from static 3D models to urban DTs therefore requires minimizing this time as much as possible during the update phase; only in this way can these models represent a significant evolution in which the urban environment is conceptualized and represented digitally, as in continuous evolution and change.

Today, despite the possibility of integration with BIM and the efforts made to develop the automatic (parametric) generation of 3D city models, which have accelerated procedures toward more accurate results and lower costs, modeling remains challenging. The difficulties are still related to the data providers and the quality of the optics and sensors used in remote sensing operations. Other aspects may be of some concern, for example, the semantic attributes and the coherence of the model elements, which must minimize the presence of errors to effectively map a complex geospatial infrastructure.

The real challenge is therefore to consider a real-time update that allows the incorporation of high levels of detail (e.g., realistic textures, geometric precision, and semantic relationships between territorial objects) [15–17]. This transition to DTs involves the need to combine real-time data from IoT sensors and process data, thanks to continuously learning artificial intelligence models. By integrating disparate data streams and analysis tools into a single, unified platform, DTs can support more holistic and integrated decision-making, leading to more informed and sustainable planning decisions [18–20].

### 3.2. An Example of Implemented Urban DT: Zurich Digitale Stadt

The above-mentioned aspects are well-included in several experiences currently underway in all major cities around the world. Certainly, a significant experience for our work on the Turin case is that of Zurich 3- to 4D (Figure 1). The geospatial dataset of the Swiss city is made freely accessible and virtually navigable through various visualization options (https://www.stadt-zuerich.ch/ted/de/index/geoz/plan-und-datenbezug/3d-stadtmodell.html, accessed on 14 April 2024). In particular, it is possible to make simple measurements in the third dimension and to simulate the projection of shadows at different times and on different days of the year. Figure 1 below shows a 3D city model and urban digital twin for the city of Zurich, Switzerland [21].

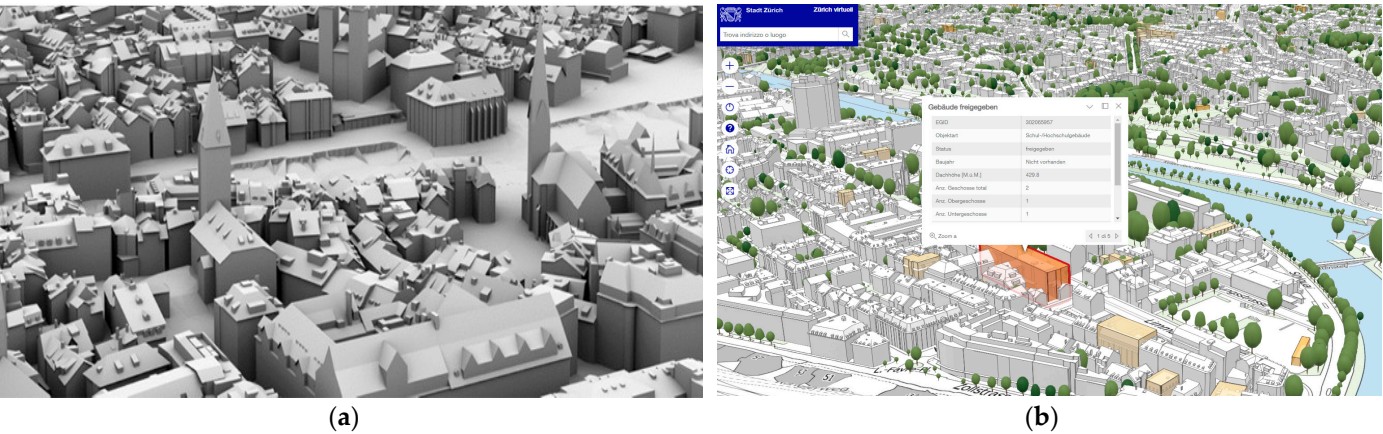

(**a**)  (**b**)

**Figure 1.** Illustration of (**a**) 3D city model vs. (**b**) 3D urban digital twin, adopted from [19]. Source: authors' reworking from [21].

The Zurich digital twin is a comprehensive spatial and digital representation of the city, including various components, such as digital terrain models, buildings, roads, and climate data. The model seamlessly integrates existing spatial data and metadata infrastructure and complies with standards such as the European INSPIRE Directive and the GeoCat 2019 catalog. Through metadata description and the use of IoT systems, real-time information on status and usage data are transmitted, enabling efficient planning and digital updates of various components within the DT, including buildings and roads. This integrated

approach to the Zurich DT facilitates the dynamic management and maintenance of urban infrastructure, increasing the city's resilience and responsiveness to evolving needs and challenges. The Zurich DT serves as a valuable tool for urban planners and policy makers, providing insight into the complex dynamics of the city and supporting informed decision-making processes. The development of the project is led by the GIS City of Zurich and has its origins in the City's Spatial Information Coordination Network with 25 service partners, which was established some 20 years ago. Open data are the foundation of the digital twin's infrastructure; from the press conference on 12 November 2018, related to the release of the 3D model of the terrain, building blocks and roofs in open format, until 20 September 2019, 11,722 datasets have been downloaded from the open data portal, a third of which correspond to the 5 datasets of the 3D spatial model of the city [21].

The experience of the Zurich DT shows that the utility of DT technology in urban planning and management lies in its dynamic and interactive nature. DTs allow for the simulation and analysis of different urban scenarios [22]. This feature is invaluable for urban resilience planning, as it allows disaster response strategies and infrastructure robustness to be tested in a virtual environment prior to real-world implementation.

In addition, DTs facilitate the optimization of urban operations through big data, such as traffic flow and energy distribution, thereby improving overall urban efficiency [23]. DT technology plays a crucial role in promoting sustainable urban development. By simulating different urban development scenarios, planners can assess the environmental impact of different initiatives and ensure that urban growth is consistent with sustainability goals [24].

It is important to recognize that certain critical factors remain present in this experience. For instance, the geospatial data utilized in the DT frequently pertains solely to the city, yet there are currently no guidelines at the cantonal or federal level that permit adequate modeling, recording, and updating. Additionally, there are no detailed methods for integrating the potential of BIM with GIS in terms of detail, automatic generalization, and mutual exchange of data. A further relevant aspect is the relationship between the quantity of data and the performance capabilities of the hardware for the construction of the 3D model, also considering the time needed to prepare them. This aspect also opens up the issue of the real-time update of the DT, which remains far from resolved. Therefore, it is necessary to understand how new point clouds from mobile mapping can be exploited to refine the quality of the 3D mesh.

### 3.3. Another Example: Helsinki 3D and Kalasatama DT

Significant experiments in urban DTs are underway in many cities around the world, often focusing on useful elements to improve urban planning, resource management, and active citizen participation in decision-making processes. In the case of Helsinki, this is one of the most significant experiments on the European scene, in which a virtual representation of the environment, together with a rich set of information services and open data, creates a dynamic model.

The 3D model, the Helsinki Reality Mesh Model (https://kartta.hel.fi/3d/mesh/, accessed on 11 March 2024), is hosted on a Cesium platform (Figure 2); the City of Helsinki has specifically experimented with the open cityGML standard in its 3D modeling work. In fact, the cityGML standard has significant advantages over static CAD models and allows urban planning and design to be much more performative; in particular, the standard allows for the integration of external data sets into the DT, thereby improving its content and visual presentation [25].

One of the most intriguing contexts for the application of the DT model in Helsinki is the district of Kalasatama (2018–2019). This area, which is situated on the outskirts of the city, has a rich history as a fishing port. However, it has undergone a significant transformation in recent years, becoming an up-and-coming seaside location. The district now boasts an industrial charm but also includes sports activities, high-rise office buildings, two city parks, and the Redi Shopping Centre. Some traditional and international restaurants have terrace tables and views of the harbor, while shops and groceries are distributed through-

out the district. A specific DT (https://kartta.hel.fi/3d/mesh/Kalasatama/, accessed on 11 March 2024) was constructed in this area with the primary objective of linking semantic data and high-resolution 3D mesh models.

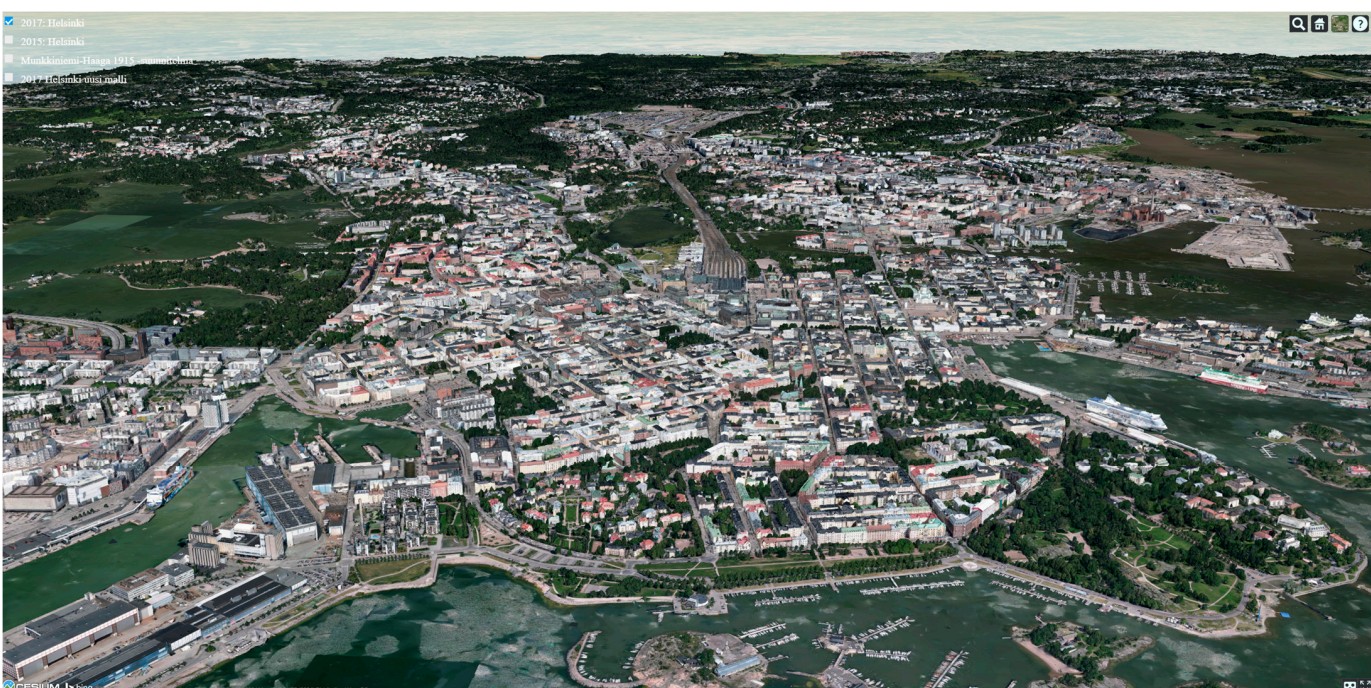

**Figure 2.** The Helsinki 3D city model (Source: adapted from https://kartta.hel.fi/3d/mesh/, accessed on 11 March 2024).

The information model of the Kalasatama district comprises three categories of buildings, including those in their current state, separately modeled buildings under construction or planned, and bridges (present and planned). Additionally, the model includes land and waterways, which are represented according to the CityGML standard. The current representation of the buildings was created in 2017 and is subject to ongoing updates. The starting data for modeling are the footprints of the buildings on the base map, which are compared to LiDAR point clouds. Other special features, such as bridges, are also included. With regard to the water bodies and the marine area, the model was created from the data on the base map and the relevant elevations.

During the construction of the DT in Kalasatama, a specific application called "Open Cities Planner" (Bentley Systems) was developed to complement and reinforce the use of the DT's platform (Figure 3). This allowed for the linking of urban regeneration data with visualizations of specific use cases. Furthermore, the Open Cities Planner application included the possibility of improving citizen participation through the proposal of surveys and the utilization of a participatory GIS (PGIS) as a basis. Another noteworthy aspect is the incorporation of wind and solar irradiance simulation data, which aims to assess the most significant air flows and examine sun-exposed areas and ground temperatures. The platform is now a model for the rest of the city to follow in future development projects and to demonstrate how digital twins can be continuously developed to meet the evolving needs of the city.

It is important to note that the Kalasatama DT experience is not without limitations. While DT has the potential to revolutionize urban design, there are still significant organizational and technical challenges that impede the widespread adoption of digital technology in the development of smart cities. This is largely due to public resistance to changing attitudes and behaviors toward the adoption of digital models. Another aspect to consider is the sheer volume and complexity of the data involved, as well as the integration between the data and the computing power needed to generate the 3D model. The generation of a

high-quality model can be laborious, as the cleaning and preparation of the same is still necessary [26].

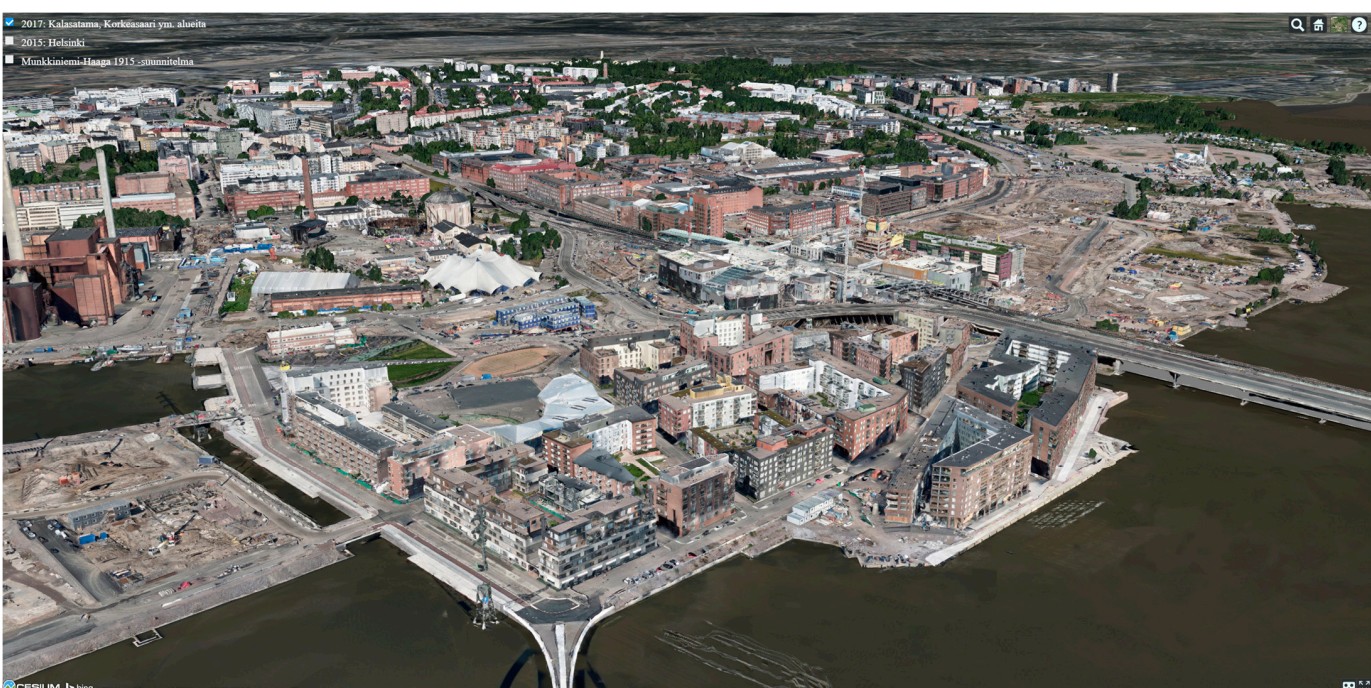

**Figure 3.** The modeled area of Kalasatama 3D mesh model (Source: adapted from https://kartta.hel.fi/3d/mesh/Kalasatama/, accessed on 11 March 2024).

### 3.4. Algorithms and Software Used for the Production of 3D City Models

The creation of 3D city models using DTs begins with the critical stage of data acquisition and processing. This process involves collecting real-world measurements from a variety of sources and then processing them so that they can be effectively used to create and update the DT. A primary method of data collection is through sensors and internet of things (IoT) devices. These devices collect a wide range of data, including traffic patterns, weather conditions, energy consumption, and more [27,28]. Remote sensing technologies, including satellite imagery and aerial photography (e.g., using drones), are crucial for capturing large-scale urban landscapes. Ref. [29] notes how these methods provide a macro-level view of urban areas. GIS plays an important role in data acquisition for 3D urban models. It involves the collection and management of geospatial data, which is essential for mapping and analyzing urban environments. In terms of data processing techniques, once the data has been collected, it needs to be cleaned and integrated. This step involves removing inaccuracies, resolving inconsistencies, and combining data from different sources into a coherent format [30]. Transforming raw data into a usable format for DTs often involves sophisticated modeling and simulation techniques. For example, [31] discusses how data are used to create detailed 3D models that accurately reflect the physical characteristics and dynamics of urban areas. Advanced data analysis plays a crucial role in interpreting the data. This can include statistical analysis, predictive modeling, and the use of machine learning algorithms to extract insights and patterns from the data [32].

The creation of 3D city models, especially when integrated with DT technology, relies on sophisticated modeling methodologies. These methodologies include a range of algorithms and software tools designed to produce accurate, detailed, and dynamic representations of urban environments. In terms of geometric and textural modeling, these are algorithms that create detailed 3D geometric representations of urban features, including buildings, roads, and natural elements. Texture modeling adds realistic surfaces to these geometries. Techniques such as photogrammetry are often used to capture real-world textures and appearances [33]. As for spatial analysis algorithms, they are usually integrated into

GIS applications and used to analyze spatial relationships and patterns in urban contexts; this is crucial for understanding urban dynamics, such as land use patterns and population distribution. Finally, simulation and forecasting algorithms can simulate urban processes (such as traffic flow or environmental changes) and predict future scenarios. These can include agent-based modeling or computational fluid dynamics [34].

Tools such as CAD and building information modeling (BIM) software (e.g., AutoDesk Auto-CAD 2024 and Revit 2022) are fundamental to the creation of detailed architectural and infrastructure models. These tools allow for the accurate modeling of individual buildings and infrastructure [35]. GIS software, such as ESRI's ArcGIS Pro or QGIS (v. 3.34), is essential for integrating spatial data into urban models. This software supports mapping, spatial analysis, and the integration of different data layers [36]. Software specifically designed for urban modeling, such as CityEngine 2023.1 or UrbanFootprint 1.5, provides tools for creating large-scale urban models. It offers functionalities for simulating urban layouts, land use, and zoning scenarios [37]. Tools such as Tableau or Power BI are used for data analysis and visualization, helping to interpret complex urban data and present it in an accessible format [38].

The 3D models with a digital representation of the physical world have also been complemented by data sharing platforms and game engines for the development of urban DTs. Game engines are the real-time simulators that have been proposed for the creation of DTs, as they have the capability of providing realistic graphics and lighting that simulate real-world conditions in a virtual replica. Modern game engines can offer various functionalities for different applications in addition to the development of game environments. Game engines such as Twin Motion, Unreal Engine, and Unity have recently been used to develop the digital twins of the city. Data platforms such as Cesium and MapStore can also provide data-sharing functionalities, as well as some basic measurement and visualization functions.

The integration of the internet of things (IoT) and artificial intelligence (AI) significantly enhances DT applications, particularly in the area of urban modeling. This integration brings a new level of intelligence and interactivity to DTs, enabling them to process vast amounts of data and provide insightful analysis for better decision-making and predictive modeling. IoT devices play a critical role in collecting real-time data from the physical environment. These data can include traffic flow, weather conditions, energy consumption and much more. IoT is essential for collecting the diverse data required for an accurate and up-to-date digital representation of the physical world. The integration of IoT devices leads to increased connectivity and interactivity within the DT. This connectivity allows for a more responsive and dynamic model that can adapt to changes in real time, increasing the usefulness of the model for urban planning and management [39]. From an AI perspective, AI algorithms are crucial for processing and analyzing the vast amounts of data collected by IoT devices. AI can identify patterns, trends, and anomalies in the data, facilitating more informed decision-making. Predictive models built using AI can forecast future urban scenarios, aiding long-term urban planning and development. AI contributes to more sophisticated decision-making capabilities within DT systems. By processing complex datasets, AI can provide insights that may not be apparent through traditional analytical methods, and AI-assisted analysis can lead to a better understanding and optimization of urban planning processes [24]. AI can also enable DTs to learn from data over time, adapting and improving their models. This aspect of machine learning allows DTs to continuously refine their accuracy and effectiveness [40].

## 4. Digital Twin Technology: Essential Elements

DT technology refers to the creation of a digital replica of a physical entity or system. This digital counterpart is not just a static representation; it is a dynamic model that mirrors the real-world object in real time [41], continuously updated with data from sensors and other sources. The core of DT technology lies in its ability to simulate, analyze, and predict behaviors of its physical counterpart, making it a powerful tool for various applications.

DTs were primarily focused on manufacturing and product lifecycle management [41]. The DT is therefore an abstraction of the structure and processes that take place in reality, a simplification of the real city. So it must be said that DT does not aim to replicate the whole real system but focuses only on some key elements and processes that describe a certain phenomenon taken into consideration at that moment; in other words, we can say that there can be a succession of models of a system that are composed partly of equal supports and partly of supports different from the basic system. However, the scope has since expanded significantly.

The evolution of DT technology over the last 20 years can be summarized as follows:

1. The concept of DT originated in the early 2000s in the context of product lifecycle management and manufacturing, as mentioned by [42]. The initial idea was to create a digital copy of a manufactured product to monitor, analyze, and optimize its performance.

2. The 2010s saw the expansion of DT technology into sectors beyond manufacturing. This was driven by advances in IoT, cloud computing, and big data analytics. In particular, the aerospace and automotive industries adopted DTs for complex system simulation and maintenance optimization [43].

3. In the late 2010s, the integration of IoT and AI technologies marked a significant evolution in the capabilities of DTs. The convergence of these technologies enabled these models to provide more comprehensive and real-time insights, enhancing decision-making processes in various industries.

4. In the 2020s, DT technology has begun to expand and advance significantly, particularly in urban planning and smart city development. By creating digital replicas of urban environments, DTs are revolutionizing urban planning, infrastructure management, and environmental monitoring. This integration greatly improves the efficiency and sustainability of urban development initiatives.

We could think of a smart city as a place where all objects with IoT connections have integrated computing and communication capabilities; data collection has therefore become relatively simple. What is complex is the convergence of the physical world with the virtual world, a convergence based precisely on the creation of a DT, i.e., a model of a physical asset that is adapted to the constant changes in the environment thanks to the use of real-time data from sensors. The same model, as mentioned above, can therefore be used to monitor and identify potential problems and anticipate their impact on its physical counterpart.

DTs can play different roles in constructing a detailed replica of the urban environment; to do so, several essential elements must be taken into account, allowing for different steps, from descriptive analysis to predictive modeling. Table 1 summarizes the three main levels of application of an urban digital twin in terms of purpose and tools.

Exploring the core components of a DT system means understanding the essential elements that provide its framework. These components are integral to the functionality, efficiency, and effectiveness of DT technology in applications ranging from industrial production to urban planning. The foundation of any DT system is robust data collection and integration. Data can include real-time information on physical conditions, operational data, and environmental inputs and is the mainstay of DT.

One of the common elements of all the current research on this topic is precisely the issue of heterogeneous data integration; this involves the collection of data from different sources, such as sensors, IoT devices, and existing databases. However, according to [44], not only are real-time data important but also the time series of data that can describe trends of past phenomena that are useful for understanding contemporary urban phenomena. Because urban phenomena are complex and often non-linear (they often give rise to wicked problems that are difficult to solve), and because urbanism is an activity that needs to delimit defined time horizons, it would be more useful to talk not only about a single DT but about several DTs [45], in which each "twin" is adapted to specific applications and different purposes. Only then should they be aggregated to form an ecosystem of DTs. Another peculiarity of DT is the continuous and bidirectional connection between the real

world and the virtual world [46], according to three steps; based on the level of integration of data from the digital and virtual worlds (Figure 4), the 3D model represents a simple abstraction of the real city.

**Table 1.** Urban DT level of application. Source: authors' work.

| Phase | Aim | Tools |
|---|---|---|
| *Analysis* | Visualize the city in 3D and its changes over time such as new assets, roads and buildings, mobility, demographic and socio-economic changes | • Overall 3D modeling<br>• 3D modeling at the street level<br>• Integration of IoT data |
| *Predictive modeling* | Model, predict, and forecast underlying activities in different sectors, such as city planning, transport, sustainability and ecology, and socio-economic trends. | • AI and machine learning to understand the relation between unstructured and structured data. |
| *Scenario planning and simulations* | Multiple what-if scenarios and simulations by pulling levers of change and its effects on the city | • Model data infrastructures: e.g., implementation of generative AI for urban planning scenarios<br>• Multi-criteria analysis for choosing the best planning scenario |

**Figure 4.** From digital model to digital twin according to the level of data integration. Source: authors' reworking from [46].

Any changes to the physical city must be manually entered into the digital copy and vice versa. A digital shadow is defined as such when data are automatically transferred from the physical city to the digital model but manually transferred from the digital model to the physical city. On the other hand, we can only consider a DT when we obtain an automatic update of the data in both directions, from real to virtual and back. In any case, the concept of urban DT represents a rather useful perspective for most cities in the world, as well as a wide field of study and research in different sectors of the academic world. Currently, looking at the characteristics of DTs already developed in different experiences, we can try to collect some common elements, even if we are far from a single and replicable standard because there is no common definition, but also because the assumptions that drive an administration to create a DT are absolutely different from city to city (Table 2). In addition, we can point out that not all cities have the same types of data (possibly open) available in terms of quantity and quality. If we look at the aspects of data integration, especially from a geospatial point of view, we can identify the peculiarities of the recent experiences of some cities.

**Table 2.** Comparison of DT characteristics in some experiences. Source: authors' reworking from [46].

| DTs | Aim | Data | 3D Model | Front-End Visualization Platform |
|---|---|---|---|---|
| *Helsinki 3D+ Kalasatama* | 3D design, urban planning, estimation of climate change, energy planning, sustainable tourism. | Oblique photogrammetry and aerial LiDAR point cloud | City Information Model, 3D CityGML model, mesh model | Semantic 3D city model: VirtualCitySystems (Berlin Germany), Cesium (Philadelphia, PA, USA), web and VR interfaces |
| *Espoo* | Urban construction, urban planning, visualization of city objects from above and underground | 3D city model database | 3D CityGML Model (textured building, generics, city furniture, water bodies, transportation, vegetation, land use and relief), 3D city model with point cloud, hybrid model, underground infrastructure | Espoo Map Service (Espoo, Finland) |
| *Vienna* | Living virtual city replica allows the monitoring of the city, generation of new information, scenario simulations, and data-driven decision support system | GIS geodata inventory and data from specialized applications from departments | Digital GeoTwin Semantics 3D geo-objects | VirtualCityMap (Berlin, Germany) |
| *Boston Digital Twin* | Digital twin helps city visualize development near beloved park. Analyzing in 3D to understand planning impact. | ESRI 3D city model | Wide variety of decision-making tasks including planning and development, flood modeling, shadow studies, and line-of-sight evaluation. | ESRI ArcGIS online (Redlands, CA, USA) |
| *Rotterdam 3D* | Climate change adaptation, viewsheds, and energy performance of buildings. Integration of the hydrodynamic city model with the 3D model | LiDAR, aerial photo, basic registration large-scale topography (BGT) and basic registration addresses and buildings (BAG), basic registration of public space. | Above and underground infrastructure BIM models and 3D city models: buildings, terrain, trees, lampposts, cables, and pipelines | VirtualCitySystems (Berlin, Germany), ESRI IMAGEM and UNITY (Redlands, USA) |
| *Zurich 3–4D* | Urban planning and climate change. Urban spatial data infrastructure | Spatial data infrastructure, geodata portal | Buildings, trees, forests, and bridges. Over 50,000 buildings in various LoD; walls and bridges | Web application, geoportal Virtual Zurich Zurich 4D (Zurich, Switzerland) |
| *Amsterdam 3D* | City planning | 3D basic addresses and buildings | Buildings, roads, vegetation, underground parts (pipelines and cables) | Unity3D (Copenaghen, The Netherlands) |
| *Virtual Singapore* | Virtual experimentation and test-bedding planning, urban planning, efficient energy consumption, population dynamics | DEM, 3D city models (buildings, roads, coastline, airspace, underground asset, and 3D geology), vegetation, cadaster, land use, water bodies, point cloud, reality mesh, BIM. | GIS data, aerial mapping, mobile street mapping, LiDAR, and imagery data. Orthophotos, CityGML used for vector models and surfaces | 3DEXPERIENCity Dassault Système (Vélizy-Villacoublay, France) |
| *Digital Twin Munich* | Climate-neutral smart cities | 3D CityGML model | Aerial surveys of the urban area, 3D point cloud mobile mapping campaigns, supplemented by drone recordings | Urban data platform based on the OGC standards (Munich, Germany) |
| *Rennes 3D* | Tackling city complexities in a systematic approach | DEM, 3D city model | 3D model demographic data relating to mobility, health, energy, vegetation | 3DEXPERIENCity Dassault Système (Vélizy-Villacoublay, France) |

**Table 2.** *Cont.*

| DTs | Aim | Data | 3D Model | Front-End Visualization Platform |
|---|---|---|---|---|
| *Virtual Gothenburg* | Urban planning, evaluation of urbanization growth, climate change affecting the sea level and posing a risk of flooding | 3D Buildings, streets, lampposts, tree plantations, and forests | Parametric, or procedural modeling | Unreal engine visualization (v. 4.27, Potomac, Maryland, MD, USA) |
| *Digital Twin Victoria* | Visualize a DT model of Victoria. Collection of 3D spatial data. | an extensive catalog of open data from across local, state, and federal government, more than 4000 datasets | Buildings, Roads, Vegetation | Virtual environment (commonly Cesium). Shared data management delivery platform Data federation approach. Open sourced TerriaJS (Philadelphia, PA, USA) |
| *Turin Digital Twin* | Urban planning and climate change. Urban spatial data infrastructure | Nadiral + Oblique photogrammetry Aerial LiDAR dense point cloud. High-resolution 3D Mesh Model | GIS data, aerial mapping, mobile street mapping, LiDAR, and imagery data. Orthophotos. | Cesium Virtual environment (Philadelphia, PA, USA). Shared data management delivery platform Data federation approach. Open-sourced MapStore (Geosolutions, Camaiore, Italy) |

## 5. The Turin Digital Twin

This section of the article focuses on the digital twins use case from the city of Turin, located in the north of Italy. The section includes the motivation, objectives of the project, progress, and challenges faced so far in the implementation of the digital twin.

### 5.1. Turin Digital Twin: Aims and Objectives

The implementation of new planning strategies to govern, predict, and manage the complexity of new urban phenomena, such as brownfields recovery, urban regeneration, climate change adaptation, and sustainable mobility, represents the future of our cities.

In Turin, as in other major European and world cities, the need to support urban planning through the creation of digital replicas of the urban environment represents an opportunity to bring together heterogeneous data and, at the same time, to integrate information within a system of structured semantic relationships between the different spatial objects.

The experiences in Turin that can be recalled so far concern the production studies of 3D models oriented toward BIM-GIS integration, the exploration of data modeling platforms at the urban scale, and their adaptability and compliance in terms of interoperability at the architectural scale [47,48]. Since 2022, the Municipality of Turin has been collaborating with the SDG11lab (https://www.dist.polito.it/en/the_department/laboratory/sdg11lab, accessed on 9 March 2024) of the Politecnico di Torino to create the digital twin of the entire city; the creation of the 3D model based on specific high-resolution aerial images is only the starting point for obtaining new geospatial data, making existing data interoperable and providing useful tools for managing urban complexity. The Turin DT project aims to contribute significantly to the development of digital technologies in the city's projects and activities and considers the applicability of DT technology in urban planning and development initiatives. The integration of existing data to make them efficient, in a perspective of advanced interoperability within the public administration, is one of the most pursued objectives; the public services of the city have, in fact, a large amount of information in the form of archives, differently conceived and managed at different times and for different purposes, often already largely digitized. With Turin DT, the intention is to rethink the existing database architectures (more geospatially oriented) with the introduction of a new metadata model and related update flows.

Agile practices of experimentation with digital technologies and services have become strategic lines of action for new urban development. In fact, the City of Turin is investing heavily in smart city projects and is trying to change the current experimentation practices in the different sectors of the city; for example, the integration of urban data, such as energy consumption of buildings or real-time traffic data, in DT models enriches the possibility of simulating the potential impact of changes related not only to the transformation of the built fabric but also those resulting from natural phenomena and meteorological conditions.

The Turin DT project aims to create an easily accessible and navigable digital city ecosystem that describes the behavior of the real world and its evolution over time, as well as the impact of future urban developments. The idea is to provide the public administration with a privileged tool for planning the future of the city while managing past, present, and future information. The construction of the 3D digital model is the elementary step to achieving these objectives; by integrating different sources and creating a virtual environment, it is possible to make explicit in real time the characteristics of the built environment, data on land use, road networks, public infrastructures, and green areas, paving the way for the real-time experience. Another final objective of the Turin DT is updating the model over time, with a planned acquisition plan and with the help of different types of sensors; this is necessary for the digital environment to maintain consistency over time, while at the same time drawing attention to a multi-temporal and multi-scale approach.

### 5.2. Data Acquisition

Turin is the third-largest city in Italy, located in the North of Italy, and is home to numerous UNESCO heritage structures. The location of the city of Turin is shown below in Figure 5.

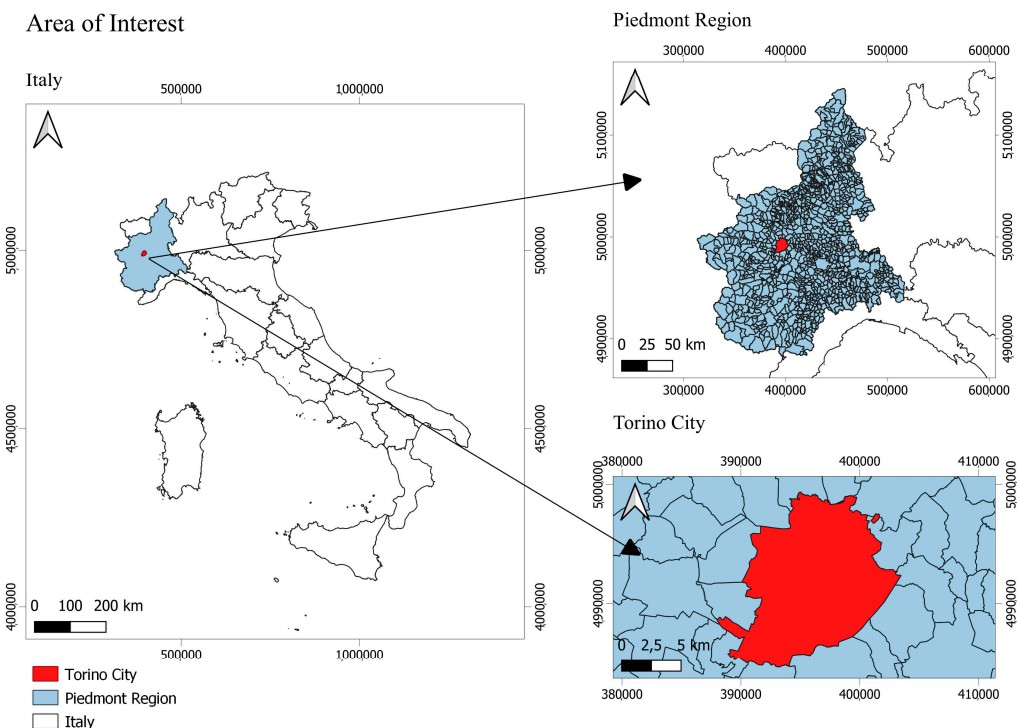

**Figure 5.** Location of the city of Turin in Italy. Source: authors' work.

The dataset used to model the Turin DT was acquired on 28–29 January 2022 using the new Leica City Mapper-2, an airborne hybrid digital sensor consisting of both optical imagery and a LiDAR point cloud (Figure 6). For the optical imagery data, 20,291 images were acquired over the city of Turin at an altitude of approximately 1 km. For each acquisition point, one nadir and four oblique images were acquired. The photogrammetric image data were characterized by a GSD of 5 cm, 60% overlap of the lateral images, and 80% overlap

of the longitudinal images. In addition, the sensor is equipped with two different cameras, including Camera NIR Lens 71 for nadir and multispectral acquisition and Camera RGB Lens 112/145 for oblique acquisition. The acquisition scheme was based on a traditional grid with nadir and oblique acquisitions. The LiDAR data were acquired simultaneously with the imagery, with a point density of 30–40/m$^2$ and an angle of 20°. This system is characterized by a conical scanning pattern, which allows vertical surfaces in all directions in the resulting point cloud.

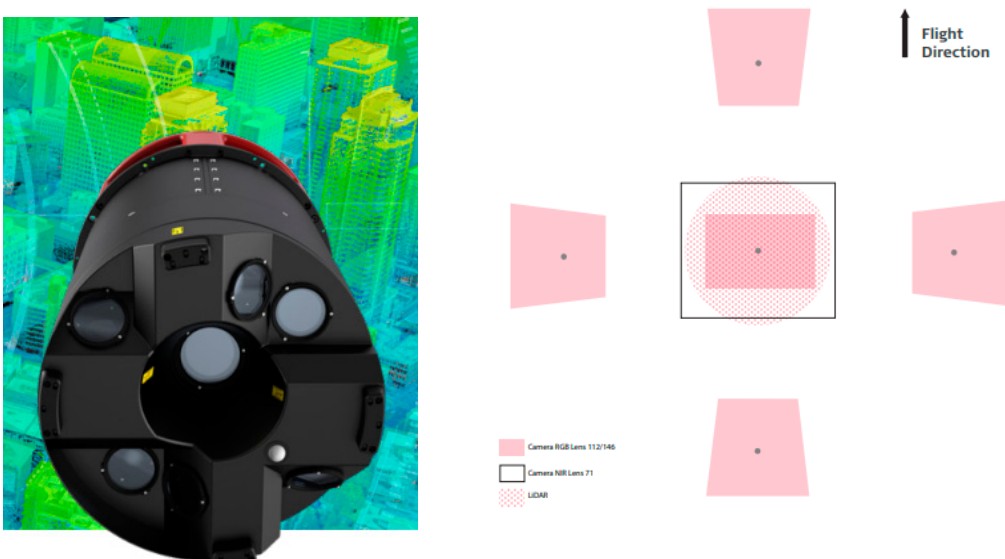

**Figure 6.** Leica CityMapper-2 system and acquisition schema used for data acquisition for Turin digital twin. Source: authors' work.

### 5.3. Data Processing Methods

The optical imagery and LiDAR data with initial orientations and trajectories were processed with Agisoft Metashape 2.1.0 and nFrames SURE 5.2 to derive classified dense point clouds, 3D meshes, accurate and detailed orthophotos, DTM, and DSM for the city of Turin (Figure 7).

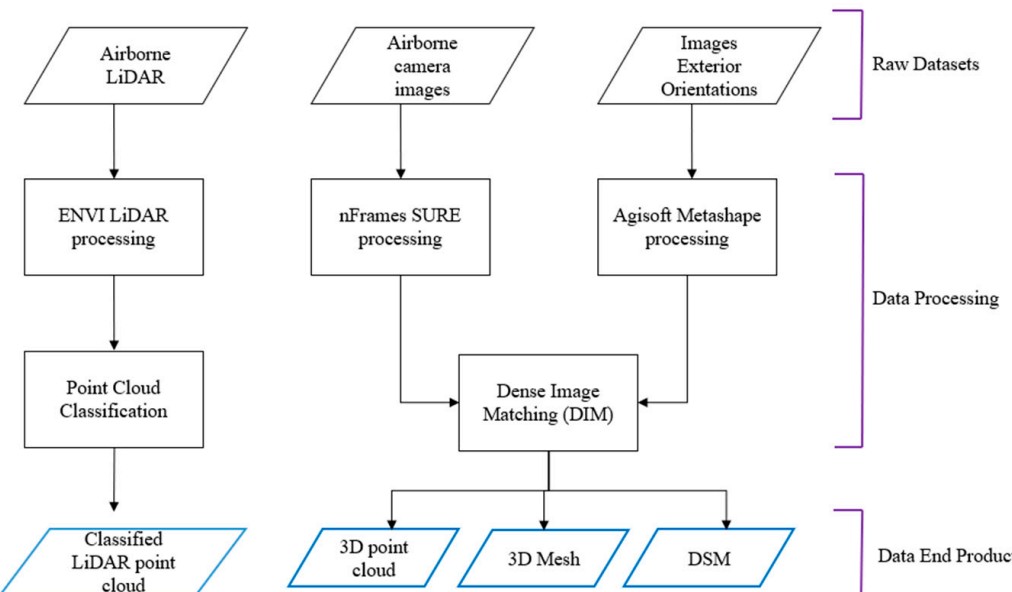

**Figure 7.** Data processing methodology. Source: authors' work.

The use of a combined dataset aims to improve the quality of the final model because the use of oblique images and the LiDAR system represent an advantage for modeling the vertical surface or provide additional information, such as intensity, which is useful for point classification. In fact, while with the image data, the classification only concerns the land use and land cover visible on the images, with the LiDAR data, we can classify the ground, vegetation, and buildings. Furthermore, a complementary benefit of using LiDAR, which is an active sensor, in combination with imagery is the ability to compensate for the need for sunlight with the ability to acquire in the shadows and under vegetation. In addition, the combination of LiDAR and imagery is essential for 3D meshing, as one is used for geometric modeling and the other for texturing the 3D model. In particular, the processing phase on nFrames SURE 5.2 combines LiDAR and image data. This software offers the possibility to use LiDAR data to improve the 2.5D and 3D products. The LiDAR point clouds are useful where the surface geometry is difficult to reconstruct from imagery alone, particularly where urban density is high or where imagery is affected by shadows or occlusion. Using the LiDAR point cloud as a complement to the image data makes the final result of the 3D model more complete and geometrically improved, as it allows for the correction of typical photogrammetric errors during the surface reconstruction phase.

### 5.4. Data Products

The 3D city model developed in the first phase of this work is the basis for the future application of the digital twin. Thanks to the synergy and integration of different technologies, the previous stages have allowed us to obtain a metrically correct model of the city, which will be useful for analyzing and describing all the different features of the original environment. For example, the 2.5D DSM makes it possible to determine the height of the building and extract the topography features, while the 3D point cloud is used to classify the points into different classes, such as building and roof shapes, high and low vegetation, and ground and road infrastructure. The 3D model can also be used for visualization and web distribution applications. All these applications can be translated into virtual layers that describe the city of Turin and can be enriched with semantic content (Figure 8).

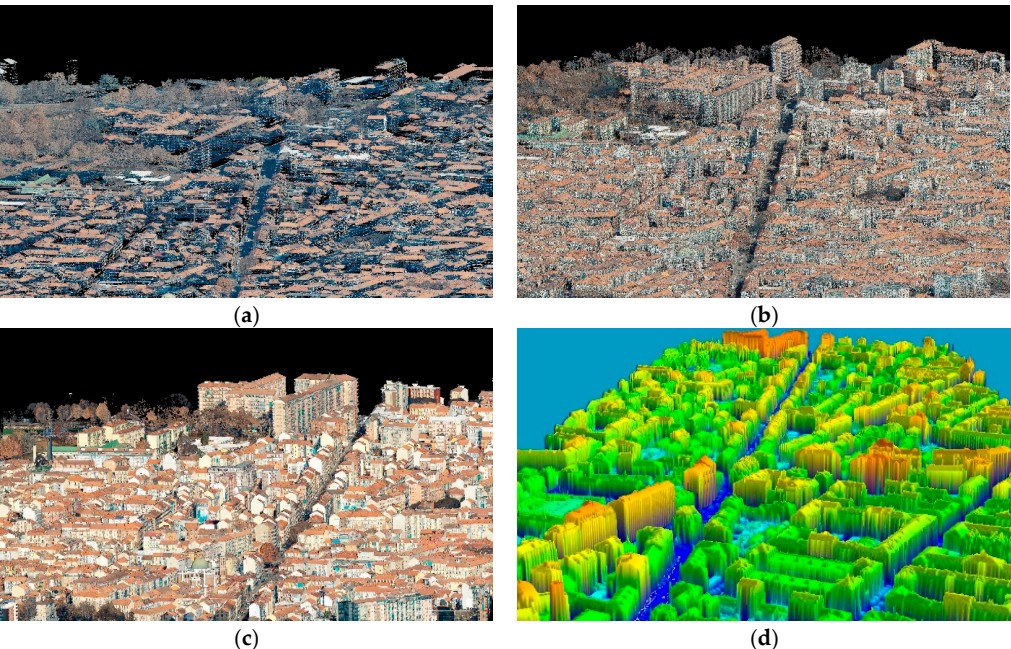

**Figure 8.** Processed data products (**a**) LiDAR point cloud; (**b**) 3D point cloud from Photogrammetry; (**c**) 3D mesh; (**d**) DSM. Source: authors' work.

*5.5. Turin DT Challenges*

The journey of developing the digital twins of Turin has been full of challenges, both technical and semantic (Figure 9). The use of 3D urban digital twins holds immense promise for urban planning, but it also comes with the complexities associated with software benchmarking, data sharing, monetary constraints, data synchronization, and data interoperability [49].

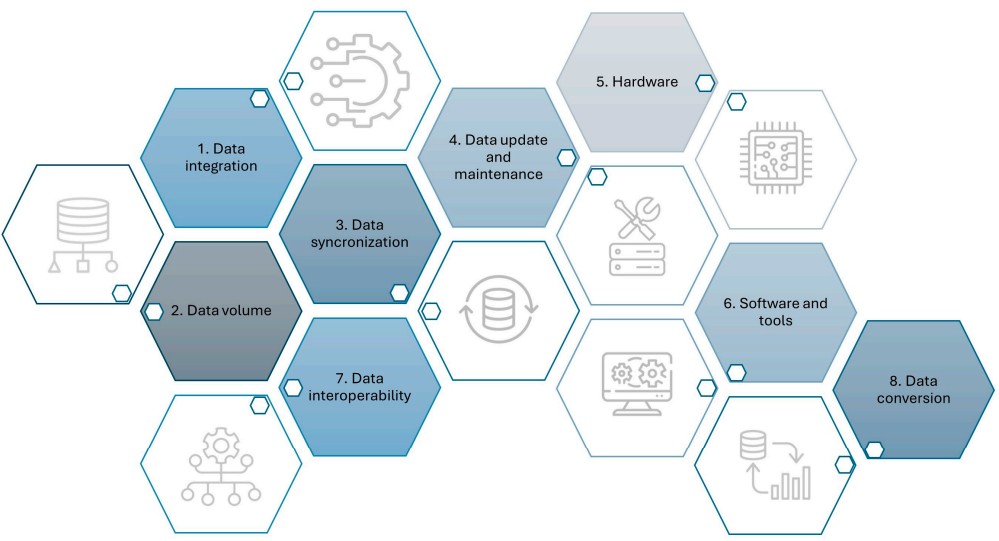

**Figure 9.** Turin DT challenges. Source: authors' work.

Software benchmarking is a crucial aspect of evaluating the performance and efficiency of different software tools for processing point cloud data and incorporating semantic information. However, the absence of standardized benchmarks and evaluation criteria makes it challenging to conduct comparative assessments and hinders progress in the field. Overcoming this challenge necessitates the establishment of rigorous benchmarking protocols to enable fair and accurate comparisons.

One of the main issues is the assignment of semantic information to the products derived from data processing and the challenges of integrating semantic information into 3D city models derived from point clouds. Without semantics, it is almost impractical to use point clouds/3D meshes/models for further analysis.

Data sharing is another major challenge, as the vast amount of data generated by urban environments requires efficient sharing mechanisms between different stakeholders. Privacy concerns, data security issues, and proprietary formats often prevent seamless data exchange, limiting collaboration and hampering the development of comprehensive urban digital twins. Overcoming this challenge requires the establishment of open data standards and protocols to facilitate interoperability and promote transparency.

Monetary constraints are a barrier to the widespread adoption of urban DTs, as the acquisition and processing of high-resolution point cloud data involves significant costs. Limited financial resources can limit access to advanced technologies, stifle innovation, and hinder progress in urban planning and management. Innovative funding models and cost-effective solutions are needed to address this challenge and encourage the wider adoption of digital twin technologies.

Data synchronization is critical to maintaining the accuracy and reliability of 3D urban digital twins, especially in dynamic urban environments in which real-time updates are essential. However, achieving synchronization between the digital model and the physical environment is technically challenging, as discrepancies in data sources and formats can lead to inconsistencies and inaccuracies. The development of robust synchronization algorithms and protocols is essential to ensure the integrity and reliability of digital twin implementations.

Data interoperability is a fundamental requirement for the seamless integration of semantic information into 3D urban digital twins, enabling disparate data sources and formats to be harmonized and effectively integrated. However, the lack of standardized data formats and protocols complicates interoperability efforts, resulting in data silos and fragmentation. The establishment of open standards and protocols for data exchange and interoperability is essential to overcome this challenge and effectively facilitate the integration of semantic information into DTs.

The Turin DT model is based on a large amount of geospatial and non-geospatial data; the redefinition of Turin as a smart city is therefore implicit, aggregating and integrating data generated by sensors and acquired both in real time and at a later date. The distinguishing feature of DT is therefore its ability to synchronize with reality, react continuously to changes in physical conditions, and provide timely indications for planning and management.

Ultimately, the integration of semantic information into 3D urban digital twins from point clouds presents several processing challenges that must be addressed to realize its full potential. By identifying the technical and semantic challenges and proposing potential solutions, the Turin DT project is progressing toward a more robust and comprehensive implementation.

## 6. Discussion

It is beyond dispute that digital twins (DTs) are powerful tools for improving decision-making, resource allocation, and urban infrastructure management. They also facilitate communication and collaboration between stakeholders and propose new urban development policies. However, the term "digital twin" is full of ambiguities. All models, to be useful, cannot reflect the totality of reality; they must necessarily be selective. City DTs are based on the latest developments of the IoT and extend the concepts of the smart city. Advanced information modeling and, more importantly, data interoperability become the key to managing the complex city. For this reason, the model must operate at two levels. At the general level, the model attempts to maintain a broad view of the entire city, including its general problems and development prospects. This is, in fact, the level of attention of a master plan. At a more "close" and local level, the specificities of places become apparent, necessitating critical and careful analysis. A number of pressing and important questions remain regarding the concept of the "frictionless future", as presented by the smart city paradigm, which envisions a seamless entity of large-scale digital infrastructure and physical cities. This vision inadvertently excludes a number of factors that are crucial to the development of urban areas. Therefore, the ambitions for digital twins of cities should ideally avoid the ambiguity and intangible nature of the smart city. To paraphrase Michael Batty [50]:

"Thus the question 'what and where is the smartest city?' not only has no answer, it is also ill-defined, largely because smartness or intelligence is a process, not an artefact or product".

This research topic presents several challenges, particularly in relation to the ongoing Turin DT project. These challenges are primarily technical and semantic in nature. From a technical standpoint, the issue of processing increasingly large and heterogeneous data sets, commonly referred to as "big data", and their cost-efficiency factor must be highlighted. From a semantic perspective, there are issues related to the speed of urban transformation and the complexity of design solutions, which increasingly require the support of the DT model. This model must be updated in real time and be navigable across multiple temporal periods.

The issue of integrating 3D city models from disparate sources to update the DT remains unresolved. An update that should be automatic is hindered by the fact that the key challenge, common to other experiences worldwide, is the lack of a simple and operational product that is easy to understand and implement "from the bottom up", directly by stakeholders.

From a technical standpoint, another challenge is the management of photogrammetric and LiDAR datasets, which often necessitate the use of increasingly powerful computing hardware for processing. When utilizing high-resolution aerial photogrammetric data (in terms of short-range), LiDAR from an airborne sensor, and the integration of other ground-based LiDAR data, it is of paramount importance to exercise caution in the georeferencing phase. In this context, it is advisable to avoid the pursuit of a single workstation with sufficient computing power. Instead, the potential offered by cloud computing (e.g., through instances on Amazon AWS, Microsoft Azure, or others) should be leveraged to reduce the cost and time of data processing.

The utilization of cloud computing can be strategically employed to facilitate the management of data, metadata, and instances that are useful for open-source visualization and sharing. In this context, the perspective of FAIR data management and sharing can present a significant opportunity for public administrations to access interactive analyses and simulations of the urban environment, as well as to facilitate collaborative and crowd-mapping solutions for updating the 3D city model.

## 7. Conclusions

The lessons learned from this project are many. From a data management perspective, the project has highlighted the importance of robust data processing capabilities and the need for significant storage and computing power. Successful implementation also requires strong collaboration between technology providers, academic institutions, and city authorities. Establishing clear data sharing and licensing agreements was essential for smooth collaboration and data distribution. In terms of benchmarking and hardware testing, selecting the right software by ensuring hardware compatibility is a critical step in projects of this scale. Benchmarking software capabilities and testing hardware configurations were key to effectively managing the massive amount of data. The Turin DT sets a precedent for future urban modeling projects. It demonstrates the potential of DTs in urban planning, infrastructure management, and emergency response planning. Future projects can leverage the methods and insights gained from the DT, potentially integrating additional data sources, such as IoT sensor data, for more dynamic and functional urban models.

The future of DT and 3D city modeling technologies is promising, with several advancements and expanding applications on the horizon, particularly in the areas of sustainability and smart cities. From a technological advancement perspective, future developments in DTs and 3D modeling are likely to see greater integration with emerging technologies such as AI, machine learning, and the internet of things (IoT). The integration of AR and VR with DT is expected to provide more immersive and interactive experiences. This could revolutionize the way urban planners and citizens engage with and understand urban models. The advent of quantum computing could offer new ways of processing the huge amounts of data involved in DTs, significantly speeding up simulations and analysis. In terms of broadening applications beyond urban planning, DTs could be used in healthcare for city-wide health monitoring and in public safety for crime prevention and response strategies, as well as in education and training, in which DTs could be used for educational purposes, providing a virtual environment for training urban planners, emergency responders, and city administrators. In terms of sustainability and smart cities, DTs are expected to play a crucial role in promoting sustainable urban development. They can help in energy optimization, waste reduction, and resource management, contributing to greener and more sustainable smart cities, enabling better urban management and improved citizen services. DTs will be instrumental in the development of smart cities, providing solutions to urban challenges through advanced data analysis and simulation [51].

The development of the DT then involves sophisticated modeling techniques that incorporate not only the physical attributes of the city but also its dynamic aspects such as traffic and population movements. The integration process is an ongoing effort, requiring regular updates and maintenance to ensure that the DT remains an accurate and reliable representation of the urban landscape. The applications of DT technology in urban plan-

ning are vast and transformative. The first primary application is in urban infrastructure management, where DTs assist in the monitoring, maintenance, and planning of urban assets. They are also crucial in urban design and development, enabling the simulation of new projects and their impact on the existing urban fabric. Furthermore, DTs play an important role in environmental management and sustainability, as they can model environmental scenarios and support decision-making for sustainable urban development [52]. Looking to the future, the potential advancements of DT technology within 3D urban modeling are promising and diverse. Future advances may include enhanced data analysis capabilities, allowing for deeper insights into urban dynamics. The integration of artificial intelligence and machine learning could enable predictive modeling and more efficient urban management. In addition, advances in immersive technologies, such as VR and AR, could transform the way stakeholders interact with and understand urban models [53].

In conclusion, the study of DTs and 3D modeling technologies in the context of urban planning and digital technology reveals several key findings with significant implications. DTs and 3D modeling technologies have emerged as transformative tools in urban planning, offering unprecedented levels of detail and interactivity. They enable more accurate and dynamic representations of urban environments, facilitating advanced scenario simulations, infrastructure management, and resource optimization. These technologies significantly enhance public engagement and policymaking processes. By providing interactive and understandable platforms, they democratize urban data, enabling more inclusive and informed public participation. For policymakers, the data-driven insights and scenario analysis provided by DTs lead to more informed and effective decision-making. Despite the benefits, challenges remain in terms of data privacy and security, technical scalability, and model accuracy. It is therefore imperative to address these challenges through the implementation of robust security measures, the utilization of advanced data management technologies, and the continuous validation of models. Furthermore, the establishment of comprehensive policy and regulatory frameworks is essential to guide the ethical and effective use of these technologies.

**Author Contributions:** Conceptualization, P.B. and L.L.R.; methodology, P.B. and L.L.R.; software, P.B., L.L.R., and Y.Y.; validation, P.B.; formal analysis, L.L.R.; investigation, L.L.R. and Y.Y.; resources, P.B.; data curation, L.L.R.; writing—original draft preparation, L.L.R.; writing—review and editing, P.B., L.L.R. and Y.Y.; visualization, L.L.R. and Y.Y.; supervision, P.B.; project administration, P.B. All authors have read and agreed to the published version of the manuscript.

**Funding:** This publication is part of the project NODES, which has received funding from the MUR—M4C2 1.5 of PNRR with grant agreement no. ECS00000036.

**Data Availability Statement:** Dataset available on request from the authors: The raw data supporting the conclusions of this article will be made available by the authors on request.

**Acknowledgments:** The authors would like to pay special thanks to their colleagues at SDG11lab (DIST Department at Politecnico di Torino) and the City of Torino, in particular to Stefano Lo Russo, Alessandra Cimadom, and Valentina Campana.

**Conflicts of Interest:** The authors declare no conflicts of interest.

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
