# Peer review of "Urban Echoes: Exploring the Dynamic Realities of Cities through Digital Twins"

_land, doi:10.3390/land13050635_

Round 1

Reviewer 1 Report (Previous Reviewer 2)

Comments and Suggestions for Authors

The article has been improved by elaborating some of the key topics. Additional references might be added in order to provide a better insight into the occurring challenges of the process application and the specificities of the listed examples. Some of them (included in the table) could be also analysed, along with the case of DT Zurich, but with a focus on the challenges and problems, as well as the benefits of the DT concept.

Comments on the Quality of English Language

The article needs just minor corrections during the final proofreading. 

Author Response

We would like to express our gratitude for the valuable review. We have endeavoured to enhance certain paragraphs and have incorporated a specific section on the Helsinki 3D case study. In our description of this case study, we have also included references to the limitations of this DT and the Zurich DT, which are in fact also the challenges that must be considered in almost all the most recent experiences. The challenges and limitations of DTs are addressed in the description of the Turin case study and in the discussion and conclusion section. Furthermore, additional bibliographical references have been incorporated, and the introductory section has been modified with a particular focus on the theme of smart city/sustainable city in relation to innovation.

Reviewer 2 Report (Previous Reviewer 3)

Comments and Suggestions for Authors

The paper has been revised, however the questions about how the Turin case got started; who initiated it, whether specific aspects of DT technology have been applied to other sectors of the Turin’s built environment, for instance, architecture; who has published about it already and more specifically what has been published about it if anything remain largely unanswered. Instead, a baseless question/answer on the smartest city was added, which likely doesn't help to advance the case for further investments in DTs and above all ignores such publications as: (i) Van Agtmael and Bakker (2016) The smartest places on earth; (ii) Gaffney and Robertson (2018) Smarter than smart; and (iii) Breznitz (2021) Innovation in real places: Strategies for prosperity in an unforgiving world. Furthermore, the opportunity to illustrate the educational and scholarly aspects associated with GIS, technology use, and community planning under Town-Gown educational collaborations appears to have been eclipsed (consult Balsas (2019) Where Does Technology Fit in the Geospatial, and (2021) GIS Buildout Analysis and Urban Planning).

Comments on the Quality of English Language

language

Author Response

We would like to express our gratitude for the valuable review. We have endeavoured to enhance certain paragraphs and have incorporated a specific section on the Helsinki 3D case study. In our description of this case study, we have also included references to the limitations of this DT and the Zurich DT, which are in fact also the challenges that must be considered in almost all the most recent experiences. The challenges and limitations of DTs are addressed in the description of the Turin case study and in the discussion and conclusion section. Furthermore, additional bibliographical references have been incorporated, and the introductory section has been modified with a particular focus on the theme of smart city/sustainable city in relation to innovation. With regard to this final aspect, we have indeed accepted the suggestions regarding the contributions of Van Agtmael and Bakker, as well as those of Breznitz, which we consider to be of great interest for the introductory part. Conversely, we have excluded Gaffney and Robertson (too specific for our purposes) and Balsas (the Buildout analysis model is interesting but does not fully align with the Italian and European planning context).

Round 2

Reviewer 2 Report (Previous Reviewer 3)

Comments and Suggestions for Authors

Grazie

Author Response

Dear Reviewer,

In this second round of reviews, we don't see any specific, timely comments that can be answered. We thank you again for the valuable comments made in the first round and we would like to inform you that we have made further changes, especially in the part concerning the challenges, and an overall review of the quality of the English language.

The Authors

This manuscript is a resubmission of an earlier submission. The following is a list of the peer review reports and author responses from that submission.

Round 1

Reviewer 1 Report

Comments and Suggestions for Authors

This is a very well-written and well-structured paper, and I commend the authors for that. However, I miss a clear research question, and a sufficiently clear, publication-worthy finding in response. The authors themselves state that "The lessons learned from this project are multifaceted," and "we focus on the ongoing progress and initiative in the [Turin] project..." But a clearer publication-worthy "takeaway" finding (or findings) is needed. Perhaps the focus could be on a specific outcome of the Turin project? Or a particular challenge of the Digital Twin methodology that is resolved in their work? 

In addition, I miss a discussion of the potential weaknesses of  the DT methodology - for example, the problem of incompleteness, "uncertainty of the data," "sensitivity to initial conditions," and related issues. Such aspects are discussed in Tzachor, A., Sabri, S., Richards, C. E., Rajabifard, A., & Acuto, M. (2022). Potential and limitations of digital twins to achieve the sustainable development goals. Nature Sustainability, 5(10), 822-829.

See also Batty, M. (2023). Digital Twins in City Planning. UCL Working Papers, Paper 237, December 2023.  The paper begins by stating "The term ‘digital twin’ is riddled with ambiguity," and it continues, "Many eloquent authors have used this example to highlight the conundrum of something which is the same (scale) as the real but intrinsically different" (p. 5). The paper quotes the mathematician and architect Lionel March, "All models – to be useful – must be selective and, if you like, false to reality in some respect or other" (p. 5).  

How is this "falseness" handled under the methodology used in the Turin project? What steps are needed to ground-truth or compensate for the inherent incompleteness of the DT? This could be discussed, perhaps as part of the publication-worthy finding (e.g. answering a research question of a similar vein).

I hope these comments are helpful, and I look forward to an improved paper that makes a substantial contribution to the literature in this important subject.   

Author Response

Dear reviewer,

The authors would like to thank the reviewers for their valuable feedback. We implemented most of your comments and, in a few cases, explained why we did not implement them. Please, see the attached file.

Kind regards

Reviewer 2 Report

Comments and Suggestions for Authors

The article, which presents a novel topic, is well structured and clearly written. The research is adequately elaborated and the discussion underlines the most important elements of the conducted study.

The Introduction part could be improved by adding the relevant references to all major themes which were mentioned and providing a brief overview/summary of the article structure.

Section 3 could be restructured by dividing the content into two sections (3.1 and 3.2), where 3.1. would provide a short comparative insight into other examples of evolving DT (beside Zurich).

Part 6 - Discussion could also include the experiences of other Italian examples (if any) or experiences of other European cities with similar characteristics.

Part 7- Conclusion - the applicability of listed recommendations to both global and local context could be elaborated.

Comments on the Quality of English Language

The article requires minor text editing. 

Author Response

Dear reviewer,

The authors would like to thank the reviewers for their valuable feedback. We implemented most of your comments and, in a few cases, explained why we did not implement them. Please, see the attached file.

Kind regards

The Authors

Reviewer 3 Report

Comments and Suggestions for Authors

The manuscript titled ‘Urban Echoes: Exploring the Dynamic Realities of Cities Through Digital Twins’ is clear and presented in a well-structured manner. The manuscript provides some background on the evolution of 3D modelling to Urban Digital Twining. While the Turin case study presents new material, the background info, advantages, limitations and current challenges are already well analysed in the literature. As such the relevance of the manuscript manifests itself in how a particular city in Italy is attempting to build its Digital Twin model based on the experiences of their counterparts mostly in Europe. The manuscript’s emphasis on ongoing work (or work in progress) is a bit disconcerting in the sense that readers are not easily able to grasp specifically what has been accomplished, the challenged encountered and how they were resolved, and what still needs to be done to ‘complete’ the next milestone(s) of the project. The paper’s aim ought to be stated more directly and the case’s scholarly significance also needs to be explained a bit more convincingly. Although the cited references are relatively recent, readers may question some the papers claims, such as: Did the use of computer science in urban planning begin with Baxter’s 1976 book? Are the authors referring mostly to Italy or to the European context? For instance, in the US, Walter Isard, Britton Harris, and Ian McHarg are well-known for their notable contributions to regional science, urban modelling, and geographic information systems, respectively. Klosterman’s paper title and year of publication are incorrect. There is another incorrect entry in “Smart Cities: Big Data, …”, which is an author’s book and not an edited volume. The manuscript is mostly descriptive and there is not enough data analysis to assess the scientificity of the case study analysis, given its ongoing nature. The manuscript could be considered more correctly as a review or a ‘progress report’ instead of a research paper. The authors are encouraged to rewrite the manuscript’s abstract, introduction, and the more descriptive passages in ways that reflect some sort of milestone that can be easily assessed and validated, for instance before the building of the digital twin continues and in order to obviate/eradicate potential mishaps. As such a well posed research question that could be adequately answered in the paper and potentially fill a certain gap in the literature would make the manuscript much more coherent and useful to other researchers (and even to the authors themselves). The manuscript’s results are not easily reproducible based on the details given in the methods section. However, the methods utilized in building the Digital Twin model thus far appear relatively plausible. The figures and tables seem appropriate, nonetheless, one wonders how the cases in Table 2 were selected for analysis. The preference for European cases and the inclusion of only two non-European cities may lead some to question why certain US cities such as Boston, MA and Phoenix, AZ were left out (see e.g. Hurtado and Gomez 2021 “Smart City Digital Twins” and Guhathakurta et al. 2009 "Digital Phoenix project”). The paper would also benefit from more evidence to prove some of its claims about DT’s potential, such as “in reducing carbon footprints, optimizing resource usage, and enhancing the overall quality of urban life” as well as “energy optimization, waste reduction, and resource management.” Could the authors back up these claims with some data and statistics? There is also no to little reference to how the Turin case got started; who initiated it? Have specific aspects of DT technology been applied to other sectors of the Turin’s built environment? For instance, architecture? Who and what has been published about it if anything? Have Town-Gown collaborations between universities / non-profit organizations and the city of Turin been productive? (see Balsas (2019) Where Does Technology Fit in the Geospatial… for an example of an environmental justice community in Upstate New York). The conclusions are a bit too generic and would benefit from greater insights from the Turin case.

Author Response

(The authors gave the same response as above.)
